# Evaluation of Serum Biomarkers and Electroencephalogram to Determine Survival Outcomes in Pediatric Post-Cardiac-Arrest Patients

**DOI:** 10.3390/children10020180

**Published:** 2023-01-18

**Authors:** Magda El-Seify, Mennatallah O. Shata, Sondos Salaheldin, Somia Bawady, Ahmed R. Rezk

**Affiliations:** 1Department of Pediatrics, Chest Unit, Ain Shams University Hospitals, Cairo 11566, Egypt; 2Department of Pediatrics, Neurology Unit, Ain Shams University Hospitals, Cairo 11566, Egypt; 3Department of Pediatrics, Ain Shams University Hospitals, Cairo 11566, Egypt; 4Department of Clinical Pathology, Ain Shams University Hospitals, Cairo 11566, Egypt; 5Department of Pediatrics, Intensive Care Unit, Ain Shams University Hospitals, Cairo 11566, Egypt

**Keywords:** brain, biomarkers, child, ischemia, neurology

## Abstract

Cardiac arrest causes primary and secondary brain injuries. We evaluated the association between neuron-specific enolase (NSE), serum S-100B (S100B), electroencephalogram (EEG) patterns, and post-cardiac arrest outcomes in pediatric patients. A prospective observational study was conducted in the pediatric intensive care unit and included 41 post-cardiac arrest patients who underwent EEG and serum sampling for NSE and S100B. The participants were aged 1 month to 18 years who experienced cardiac arrest and underwent CPR after a sustained return of spontaneous circulation for ≥48 h. Approximately 19.5% (*n* = 8) of patients survived until ICU discharge. Convulsions and sepsis were significantly associated with higher mortality (relative risk: 1.33 [95% CI = 1.09–1.6] and 1.99 [95% CI = 0.8–4.7], respectively). Serum NSE and S100B levels were not statistically associated with the outcome (*p* = 0.278 and 0.693, respectively). NSE levels were positively correlated with the duration of CPR. EEG patterns were significantly associated with the outcome (*p* = 0.01). Non-epileptogenic EEG activity was associated with the highest survival rate. Post-cardiac arrest syndrome is a serious condition with a high mortality rate. Management of sepsis and convulsions affects prognosis. We believe that NSE and S100B may have no benefit in survival evaluation. EEG can be considered for post-cardiac arrest patients.

## 1. Introduction

Post-cardiac-arrest syndrome, which includes whole-body ischemia and subsequent reperfusion, is a common occurrence after the return of spontaneous circulation (ROSC) [1]. The pathological effects of this syndrome impair the myocardial function with systemic ischemia–perfusion and cause brain injury [2].

Cardiac arrest can cause primary and secondary brain insults. Nonreversible primary insult occurs at the time of arrest, and a reversible secondary injury may occur following ROSC and subsequent cerebral reperfusion [3]. According to the American Heart Association, nearly 6000 infants and children develop in-hospital cardiac arrest (IHCA) annually, with 89% of survivors having favorable neurological outcomes [2]. The estimated survival rates for IHCA range from 16% to 38% in different studies [4]. Age may affect the survival rate in pediatric intensive care unit (ICU) patients; furthermore, neonates and infants have better survival than older children [5]. The elderly have a lower thirty-day survival than younger patients [6].

For the detection and quantification of the magnitude of brain insult as well as evaluation of the efficacy of therapeutic strategies and outcome prediction, serum brain-specific biomarkers, such as neuron-specific enolase (NSE) and serum S-100B (S100B), can be used [7]. S100B, a calcium binder, is a homodimer protein in glia and Schwann cells that regulates calcium-dependent cellular signaling in neuronal differentiation, outgrowth, and apoptosis through different concentrations. It can augment brain damage by inducing the release of inflammatory cytokines. NSE is a gamma isomer of enolase and a cytoplasmic enzyme of glycolysis. When brain damage occurs, it is released into the bloodstream and cerebrospinal fluid due to disruption of the blood–brain barrier [8].

Electroencephalogram (EEG) is a bedside tool that has been used to evaluate the degree of coma and the severity of damage following cardiac arrest [9]. Many EEG patterns have been associated with poor functional outcomes; the most reliable are generalized suppression of <20 μV, burst suppression pattern with generalized epileptiform activity, and generalized periodic complexes on a flat background [10].

To the best of our knowledge, the incidence of post-cardiac-arrest syndrome in children and associated mortality have not been evaluated in Egypt and the Middle East. In the present study, we evaluated the correlation between NSE, serum S100B, and EEG patterns in pediatric patients with in-hospital cardiac arrest circumstances and assessed patient outcomes, including the duration of CPR and survival.

## 2. Materials and Methods

This prospective observational study included 41 post-cardiac-arrest pediatric patients (24 [58.5%] males and 17 [41.5 %] females) admitted to the pediatric ICU of our hospital from January 2017 to December 2019. The median age was 6 months (range: 3 months to 12 years); approximately 56% (*n* = 23) of the participants were infants (1 month–<1 year); 27% (*n* = 11) were young children (1 year–<8 years), and 17% (*n* = 7) were older children (8–12 years).

Patients aged one month to 18 years who suffered cardiac arrest and underwent cardiopulmonary resuscitation (CPR) followed by a sustained ROSC for ≥48 h were included in this study. Patients who were aged <1 month or >18 years; those with neurological diseases (e.g., cerebral palsy, neurodegenerative disease, and encephalitis), hematological malignancies or solid tumors; those who were immunocompromised; and/or those who had a history of head trauma were excluded from this study.

All patients in the study group were subjected to data collection as per the Utstein reporting guidelines for IHCAs [11]. Data regarding CPR were collected and recorded by interviewing the CPR providers within 24 h of arrest and reviewing medical records [12].

Patient variables that were collected included age, sex, order of birth, consanguinity, originally affected systems, date of ICU admission, invasive mechanical ventilation before the arrest, and cardiac support (dopamine, dobutamine, adrenaline, and noradrenaline) before the arrest; all were reported using the official hospital records of the patients.

Event variables were the immediate cause of the arrest and the duration of CPR. Outcome variables were survival to ICU discharge, mortality within 5 days and after 5 days of the cardiac arrest, convulsions, the need for sedation or anticonvulsants, and associated sepsis within 48 h after cardiac arrest, as evidenced by laboratory data including complete blood count (CBC), C-reactive protein (CRP) level, and positive blood culture. Approximately 2 mL of fresh venous blood was collected for CBC in a tube containing ethylenediaminetetraacetic acid as an anticoagulant. CBC was performed using Sysmex XT-1800i (Sysmex, Kobe, Japan). CRP level was measured using the semi-quantitative latex agglutination test (Avitex CRPkit, Omega Diagnostic Limited, Scotland, United Kingdom). Serum electrolytes Na^+^ and K^+^, blood urea nitrogen, creatinine, and alanine aminotransferase levels were also measured. Blood samples for serum brain-specific biomarkers were withdrawn 48 h after admission. We considered the first 48 h to correlate with the half-life of NSE (i.e., 24–96 h) and S100B (i.e., 0.5 h) and to avoid new-onset infections or any given treatment that may affect these markers’ levels [13,14].

NSE assessment was performed using the Cobas e411 (Roche Diagnostics Ltd., Rotkreuz, Switzerland) fully automated system, and S100B assessment was performed using a double-antibody sandwich enzyme-linked immunosorbent assay. Further explanation is provided at the Appendix A. EEG was performed during the first 48 h post-cardiac-arrest using Nicolet REF 515-019000 rev 06 (CareFusion, Middletown, WI, USA), and the recorded results were interpreted by a neurology consultant. Brain-damage diagnosis was confirmed through clinical examination done by a neurology consultant.

### Statistical Analysis

The collected data were revised, coded, tabulated, and computed using IBM SPSS Statistics version 23.0 (IBM Corp., Armonk, NY, USA) using appropriate statistical methods. Data were presented as median (interquartile range) if skewed. Categorical variables were compared using Fisher’s exact test. Power analysis was performed before patient recruitment to validate that the sample size was sufficient to support the statistical significance of the study outcomes with a confidence interval of 95% and a 5% margin of error. A *p* value of ≤ 0.05 was considered statistically significant. The diagnostic or predictive values of NSE for predicting cardiac-arrest duration (CPR duration) of more or less than 7 min were determined using a receiver-operating characteristic (ROC) curve, and the area under the ROC curve (AUC) was interpreted according to Kumar and Indrayan [15].

## 3. Results

The study included 41 post-cardiac-arrest pediatric patients; the cause of arrest was divided into hypoxic arrest (respiratory arrest) and cardiac arrest. Hypoxic arrest was more common than cardiac arrest (80.5%, *n* = 33). The median duration of CPR was 7 min (range: 1–20 min). Regarding survival, older and young children (28.57% and 27.27%, respectively) demonstrated higher survival rates than infants (13%). The overall survival was 19.5%. As regard patients diagnosed with sepsis, 60% of blood cultures growth were klebsiella pneumoniae while the rest were coagulase-negative staphylococci (Table 1).

Convulsions and sepsis after cardiac arrest were common features noted in 21.9% and 82.9% of the participants, respectively.

To assess brain damage, serum NSE and serum S100B were recorded, and the EEG findings were noted after the cardiac-arrest event. Table 2 and Table 3 demonstrate that the difference in the values of NSE and S100B levels between patients who died and those who survived were not statistically significant (*p* value 0.2 and 0.3 respectively). However, the EEG results were significantly associated with survival. Non-epileptogenic and generalized epileptogenic activity patterns were associated with the highest survival rate (36.8% survived). A burst suppression pattern was associated with the lowest survival duration (all patients died within 5 days 0%suvival) and the lowest median survival duration after arrest (2 days) (Table 2 and Table 3). Generalized epileptogenic activity presented as generalized tonic–clonic seizures.

Figure 1 and Figure 2 demonstrate that a statistically significant positive correlation existed between serum NSE level and the duration of CPR (R = 0.339, *p* = 0.03), whereas no correlation was found between S100B level and the duration of CPR (R = −0.11, *p* = 0.493).

NSE revealed good predictive or logistic value for brain-damage diagnosis, with an AUC of 0.72. The cut-off value of serum NSE (3.5 ng/mL) correlated with CPR duration of more than 7 min, with a sensitivity of 95.45 % and specificity of 52.6% (Figure 3).

Notes: Cut-off Point, ˃3.5; AUC, 0.72; Sensitivity, 95.45; Specificity, 52.63; PPV, 70; NPV, 90.9.Abbreviations: AUC, area under the ROC curve; NPV, negative predictive value; PPV, positive predictive value; ROC, receiver-operating characteristic.

Sepsis and convulsions were associated with an increased risk of mortality (relative risk of 1.99 [95% CI, 0.8–4.73] and 1.33 [95% CI, 1.09–1.6]), respectively, as illustrated in Table 4.

## 4. Discussion

Cardiac arrest is an emergency condition among both hospitalized and non-hospitalized children. Sequelae can follow cardiac arrest that pediatricians and the patients’ families must face and deal with. Studying the conditions and survival rates of patients after cardiac arrest is an attempt to identify a useful tool to predict outcomes and improve CPR techniques.

This study assessed the survival rates and outcomes of patients who had a cardiac arrest and the return of circulation after CPR, using EEG and two serum biomarkers. Approximately 19.5% of pediatric patients with IHCA survived until ICU discharge. Similar studies have reported survival rates of 35%, 34.8%, and 45% [16,17,18]. Surprisingly, infants in our study were less likely to survive until ICU discharge (13%) than young (1 year–<8 years) (27.27%) and older children (8–12 years) (28.57%). This finding contrasted with findings by Jayaram et al., who reported a higher survival rate among infants (30.8%) than young children (22.1%) and old children (16.8%) [16]. This difference might be explained by the small number of patients and the large proportion of infants in our study group.

Convulsions are common in post-cardiac-arrest patients (21.9% in our study and 25.7% of patients in a study by Topjian et al.) [19]. In both studies, mortality was higher in patients with convulsions. The poor outcome in patients with convulsions could be explained by the fact that convulsions could be a marker of the severity of the initial brain insult. Moreover, convulsions increase the metabolic demand of the brain, causing further neuronal injury [20].

Sepsis was diagnosed in 82.9% of patients in our study cohort and was associated with high mortality (relative risk of 1.99). Only 14.7% of patients with sepsis survived ICU discharge. A high prevalence of sepsis in pediatric patients with cardiac arrest has also been observed in single-institution and multicenter registry-based pediatric studies [21]. Sepsis was identified in 9–48% of cases in some single-institution studies and ranged from 14% to 34% in multicenter studies [21]. Septic patients have worse outcomes [22,23]. Pediatric data from the resuscitation database demonstrated odds of survival to discharge of 0.65 among children with cardiac arrest associated with sepsis [16]. The multinational Euromerican pediatric cardiac-arrest study network found that mortality was 78.8% in patients with sepsis; the relative risk of mortality was 2.64 higher for children with sepsis compared with those without sepsis [23]. In a study by Coba et al., bacteremia, identified by positive blood culture, was studied in 173 post-cardiac-arrest adults. Bacteremia was present in 38% of patients in the study group, and the mortality in the emergency department was significantly higher in the bacteremia group (75.4%) than in the non-bacteremia group (60.2%), with a *p*-value < 0.05 [24].

Our NSE and S100B levels differed from those reported in a study by Fink et al. [7], wherein serum biomarker concentrations were measured at several time points between 0 and 120 h after ROSC. Children with cardiac arrest whose biomarker levels were within the normal range demonstrated favorable outcomes. In contrast, patients who died had noticeably higher NSE and S100B levels at 24 h. The concentration of NSE and S100B at 48 and 72 h post-ROSC significantly increased in participants who died in contrast to what was observed in participants who survived. According to Topjian et al., survival could be predicted by the S100B levels measured at 48 and 72 h [19]. Moreover, there was an association between all-time points and neurological outcome and survival in a study by Fink et al. [7].

Our results are similar to those of a study by Song et al., wherein S100B level was measured twice before starting CPR (first S100B) and immediately after ROSC (second S100B) [25]. Song et al. demonstrated no association between serum S100B levels and the long-term outcomes of cardiac arrest. Thus, brain ischemia or any other extra-cranial origin may be the cause of S100B elevation in cardiac arrest [26]. Furthermore, concerning the timing of S100B release, previous studies measured S100B levels after ROSC, within 24 h or after 1 day, and presented a notable association between S100B levels with long-term outcomes and neurologic function. Regarding the difference in S100B levels between survivors and non-survivors at admission, this study did not demonstrate a significant difference between the two [26]. However, our study is a single-center study with a small sample size, making it difficult to generalize the conclusion. Moreover, we focused on the level of S100B at a single time point and did not follow up with the levels at different time points.

A change in serum biomarker levels could indicate an ongoing brain insult and influence survival. As previous studies included follow-up periods of 24 h or more to measure serum brain-specific biomarkers, the limited role of biomarkers in this study should be cautiously evaluated. Moreover, the non-association between overall survival and biomarker levels in our study could be explained by the fact that the mortality of post-cardiac-arrest patients was not mainly dependent on brain insult. We hypothesize that mortality was more commonly caused by circulatory or respiratory failure, which is supported by the high percentage of sepsis associated with multi-organ failure. In addition, a long duration of dependency on respiratory and cardiac support in all our post-cardiac-arrest patients suggests the marked compromise of the cardiorespiratory system. Therefore, we recommend further studies on the markers of cardiac and respiratory failure and on sepsis scores as prognostic factors for predicting survival in pediatric post-cardiac-arrest patients.

The absence of epileptogenic activity pattern had the highest survival rate, as 36.84% of patients survived and had the longest survival duration after arrest (median of 6 days). Burst suppression demonstrated no survival (0%) and the lowest survival duration after arrest (median of 2 days). Moreover, EEG recordings demonstrated no epileptogenic activity in 75% of survivors. Thus, EEG findings were significantly associated with mortality (*p* = 0.01). Kaplan–Meier analysis of the relationship between the EEG and overall survival was highly statistically significant, with a *p*-value of 0.001.

In our study, 53.1% of patients with no convulsions had abnormal EEG findings. Abnormal EEG findings have different prognoses and require specific treatment, which is why we recommend EEG to all post-cardiac-arrest patients. Nishisaki et al. developed an EEG grading system as follows: grade 1, continuous, not low voltage or slow; grade 2, continuous, low voltage, or slow; grade 3, continuous, low voltage, and slow; grade 4, discontinuous; grade 5, isoelectric [27]. There was a relationship between EEG patterns and outcomes in children following cardiac arrest. Background slowing and low voltage, discontinuous EEG, and isoelectric EEG were found to be the poorest prognostic factors for neurologic status at discharge. Good neurological outcomes have been reported in 90.9% of patients with grade 1 or 2 EEG recordings compared with 54.6% of patients with grade 3 and 10% of patients with grades 4 or 5 [27]. In a study by Zandbergen et al., 276 EEGs were performed 72 h after CPR in adults, and a burst suppression pattern was found to be an indicator of poor neurological consequences [28].

This study is limited by the short follow-up duration and the small sample size in a single center; therefore, further research is required.

In conclusion, cardiac arrest is one of the greatest emergencies. Sepsis and convulsions are associated with high mortality rates. We presume that NSE levels correlated with CPR duration, not overall survival, while serum S100B levels were not related to CPR duration nor overall survival. Different EEG patterns were related to post-cardiac-arrest outcomes. We recommend further studies on the markers of cardiac and respiratory failure and on sepsis scores as prognostic factors for survival in pediatric post-cardiac-arrest patients. We also suggest that EEG should be performed for all post-cardiac-arrest patients.

## Figures and Tables

**Figure 1 children-10-00180-f001:**
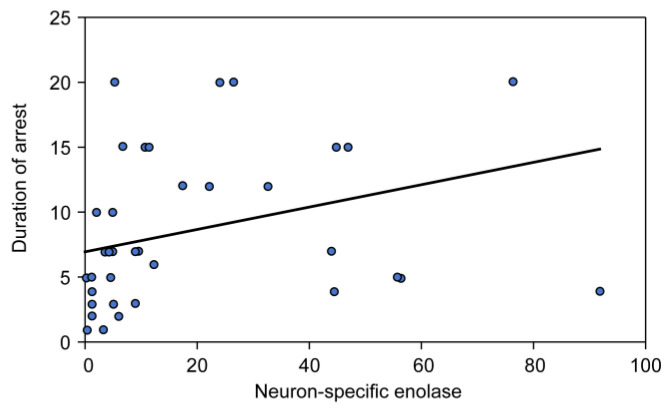
Correlation between neuron-specific enolase and the duration of CPR.

**Figure 2 children-10-00180-f002:**
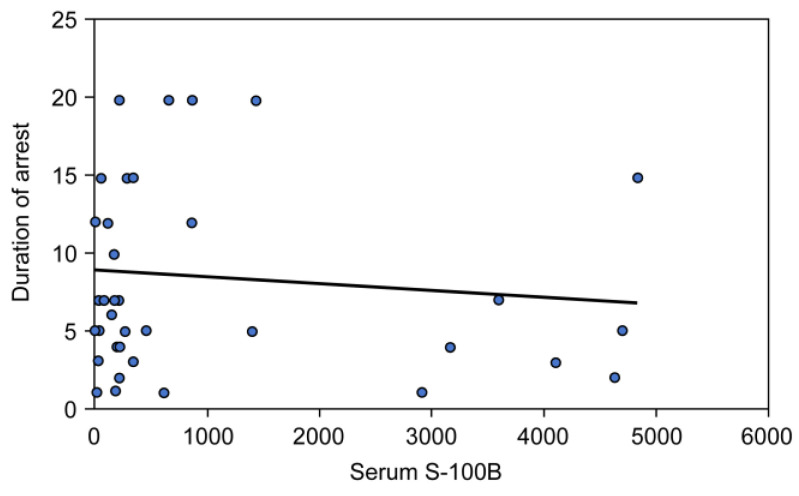
Correlation between serum S-100B and the duration of CPR.

**Figure 3 children-10-00180-f003:**
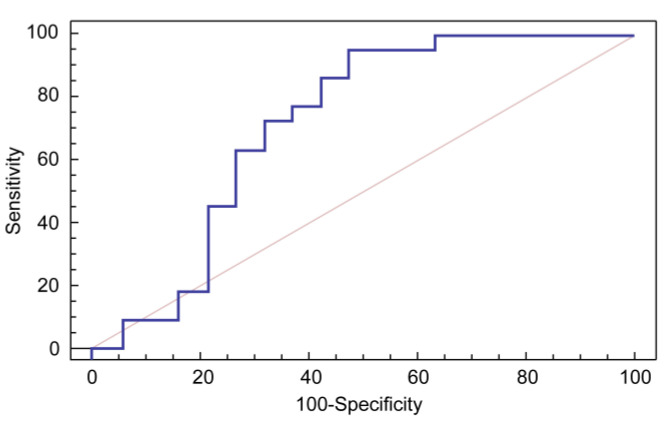
ROC curve for sensitivity and specificity of neuron-specific enolase for predicting the duration of CPR.

**Table 1 children-10-00180-t001:** Serum biomarkers and outcomes of cardiac arrest in the study group.

		Patients (*n* = 41)
Cause of arrest	Hypoxic	33 (80.5%)
	Cardiac	8 (19.5%)
Duration of arrest (min)	Range	1–20
	Median (IQR) ^‡^	7 (4–15)
Mortality	Died within 5 d	12 (29.3%)
	Died after 5 d	21 (51.2%)
	Survived	8 (19.5%)
Convulsions	Negative	32 (78.0%)
	Positive	8 (19.5%)
	Status epileptics	1 (2.4%)
Sepsis	Negative	7 (17.1%)
	Positive	34 (82.9%)
Neuron-specific enolase (ng/mL)	Range	0.5–91.88
	Median (IQR)	6.78 (3.5–26.64)
Serum S100B (ng/mL)	Range	0.00001–4.833
	Median (IQR)	0.219 (0.0954–0.8664)
EEG ^†^	No epileptogenic activity	19 (46.3%)
	Generalized epileptogenic activity	12 (29.3%)
	Focal epileptogenic activity	2 (4.9%)
	Suppressed background	4 (9.8%)
	Burst suppression	4 (9.8%)

Abbreviations: ^†^ EEG, electroencephalogram; ^‡^ IQR, interquartile range.

**Table 2 children-10-00180-t002:** Relationship between biomarkers, EEG, and outcomes of the studied patients.

		Outcome			Test Value	*p*-Value
		Died within 5 d (*n* = 11)	Died after 5 d (*n* = 21)	Survived (*n* = 9)		
Neuron-specific enolase	Range	1.5–76.5	0.5–76.5	0.61–91.88	2.558	(0.278)
	Median (IQR) ^‡^	12.55 (5.47–44.11)	6.7 (3.3–26.64)	5.31 (3.49–9.18)		
Serum S100B	Range	0.0263–3.6045	0.00001–4.7015	0.00001–4.8333	0.734	(0.693)
	Median (IQR)	0.1732 (0.0954–1.4)	0.2601 (0.091–0.6011)	0.3264 (0.1754–3.1667)		
EEG ^†^	No epileptogenic activity	3 (27.3%)	9 (42.9%)	7 (77.8%)	19.806	(0.011)
	Generalized epileptogenic activity	2 (18.2%)	9 (42.9%)	1 (11.1%)		
	Focal epileptogenic activity	0 (0.0%)	2 (9.5%)	0 (0.0%)		
	Suppressed background	2 (18.2%)	1 (4.8%)	1 (11.1%)		
	Burst suppression	4 (36.4%)	0 (0.0%)	0 (0.0%)		

Abbreviations: ^†^ EEG, electroencephalogram; ^‡^ IQR, interquartile range.

**Table 3 children-10-00180-t003:** Kaplan–Meier analysis for the relationship between the studied markers and overall survival.

		No.	^‡^ OS (d)		95% Confidence Interval		Log Rank Test	
			Median	^§^ SE	Lower	Upper	*X^2^*	*p*-Value
Neuron-specific enolase	≤6.78	16	6	1.984	2.111	9.889	1.593	0.207
	>6.78	17	4	0.823	2.387	5.613		
Serum S100B	≤0.2193	19	6	0.526	4.97	7.03	0.926	0.336
	>0.2193	14	4	0.598	2.829	5.171		
EEG ^†^	No epileptogenic activity	13	6	1.169	3.708	8.292	17.913	0.001
	Generalized epileptogenic activity	11	6	1.651	2.763	9.237		
	Focal epileptogenic activity	2	5					
	Suppressed background	3	4					
	Burst suppression	4	2	0.433	1.151	2.849		

Abbreviations: ^†^ EEG, electroencephalogram; ^‡^ OS, overall survival; ^§^ SE, standard error.

**Table 4 children-10-00180-t004:** Relationship between the presence of sepsis and convulsions and mortality.

*p*-Value *	Survived	Died	
0.03	5 (14.71%)	29 (85.29%)	Septic patients (*n* = 34)
	4 (57.14%)	3 (42.86%)	Non-septic patients (*n* = 7)
0.164	0 (0%)	9 (100%)	Convulsions (*n* = 9)
	8 (25%)	24 (75%)	No convulsions (*n* = 32)

* Fisher’s exact test.

## Data Availability

The data presented in this study are available on request from the corresponding author. The data are not publicly available due to other unpublished articles based on this database.

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
