# Peer review of "Evaluation of Serum Biomarkers and Electroencephalogram to Determine Survival Outcomes in Pediatric Post-Cardiac-Arrest Patients"

_children, 2023, doi:10.3390/children10020180_

Round 1
Reviewer 1 Report
This study examines the ability of serum biomarkers, specifically NSE and serum S-100B to predict outcomes in pediatric patients with in hospital cardiac arrest. It also evaluates the relationship between specific EEG patterns and outcomes in this patient population. There was no significant relationship between serum biomarkers and patient outcomes, however certain EEG patterns are related to worse outcomes.
Comment 1: In Table 1, please add units to the table (ie range of duration (minutes)).
Comment 2: Please include which bacteria were identified in the patients with positive blood cultures. Did any of the patients have a lumbar puncture to identify potential meningitis/encephalitis, or was this not clinically evident?
Comment 3: What specifically were the generalized epileptogenic activity patterns identified? ex. diffuse spike-wave generalized seizures, myoclonic seizures, tonic seizures, generalized periodic discharges etc. Was the percent time spent in these patterns identified over the 48 hours of recording?
Comment 4: How many patients with seizures on EEG had associated clinical activity vs how many were electrographic? Was this associated with outcomes?
Comment 5: Did any of the patients have any other coma patterns on EEG? ie alpha coma, beta coma, spindle coma
Comment 6: Line 239 refers to an EEG background grading scale. Was this done in this population in the study? Was EEG reactivity tested?
Comment 7: Was the burst suppression ratio calculated for patients/ did this have any significant relationship with outcome?
Comment 8: What were the MRI findings on these patients, if performed?
Comment 9: Line 147 refers to 'brain damage diagnosis'. What was used to make this diagnosis?
Reviewer 2 Report
I commend the researchers for collecting biosamples in such a vulnerable cohort! It is not an easy task.
1. The conclusions are too bold for a small single-center study "NSE and S100B have no benefit in survival evaluation. EEG is recommended for all post-cardiac arrest patients." Especially because it significantly conflicts with Fink et al. results recently published in JAMA.
2. Why did they pick 48 hours? Lacking an initial score and/or longitudinal measurements significantly weakens this study.
3. How were the biomarkers collected and processed?
4. I suggest making it very clear that this study refers to IHCA and not OHCA. There is a significant difference.
5. I think more than a correlation between NSE and S100B and the duration of cardiac arrest, the more interesting question here is do the levels correlate with patient outcomes. The reason is that that this finding is not novel. We know from other studies that the longer the CPR, the higher the damage and the biomarker levels.
Round 2
Reviewer 2 Report
Unfortunately, the revision is not sufficient. The abstract still says, "NSE and S100B have no benefit in survival evaluation. EEG is recommended for all post-cardiac arrest patients." This is too bold of a conclusion for a single-center study.
I still do not understand why they picked 48 hours, based on what scientific ground? And still believe is hard to make conclusions without any longitudinal measurements or outcomes. The authors have not provided a compelling answer.
Moreover, we need to know how the biomarkers were processed exactly. This is not an acceptable answer for a scientific paper "Biomarkers were collected under complete sterile conditions through central venous catheter. What I mean by this is the exact scientific methodology, i.e., common data elements biomarker collection in TBI (which tubes, what temperature, how long were they centrifuged, for how long were they frozen, etc.) This is to determine the reproducibility of this study.
Please make sure to indicate which line number for the corrections.
